# Model evaluation by a cloud classification based on multi-sensor observations

Akio Hansen[1,2], Felix Ament[1,2], Verena Grützun[1], and Andrea Lammert[1]

[1]Meteorological Institute, University Hamburg, Bundesstraße 55, 20146 Hamburg, Germany
[2]Max-Planck-Institute for Meteorology, Bundesstraße 53, 20146 Hamburg, Germany

**Correspondence:** Akio Hansen (akio.hansen@uni-hamburg.de)

**Abstract.** The detailed understanding of clouds and their macrophysical properties is crucial to reduce uncertainties of cloud feedbacks and related processes in current climate and weather prediction models. Comprehensive evaluation of cloud characteristics using observations is the first step towards any improvement.

An advanced observational product was developed by the Cloudnet project. A multi-sensor synergy of active and passive remote-sensing instruments is used to generate a Target Classification providing detailed information about cloud phase and structure. Nevertheless, this valuable product is only available for observations and there is yet no comparable surrogate for models. Therefore, a new cloud classification algorithm is presented to calculate a comparable classification for models by using the temperature, dew point and all hydrometeor profiles.

The study explains the algorithm and shows possible evaluation methods making use of the new synthetic cloud classification. For example, the statistics of the vertical cloud distribution as well as e.g. the accuracy of cloud forecasts can be investigated regarding different cloud types. The algorithm and methods are exemplarily tested on two months of operational weather forecast data of the COSMO-DE model and compared to a Cloudnet supersite in Germany. Additionally, the cloud classification is applied to Large Eddy Simulations with a similar resolution as of the observations showing detailed cloud structures.

## 1 Introduction

Clouds and their related processes are still responsible for the highest uncertainties of current climate and weather prediction models (IPCC, 2007; Forster et al., 2007). They are of great importance for accurate weather predictions to various end users and applications like solar power forecasts (Huang and Thatcher, 2017; Antonanzas et al., 2016; Sperati et al., 2016), the aviation sector (Bolgiani et al., 2018; Gultepe et al., 2015) and many more. Nevertheless, the evaluation of the macrophysical cloud properties of current atmospheric models is even nowadays very challenging due to the complexity of involved processes and the large variability of clouds.

The Cloudnet project, started in 2001, established a framework of continuous cloud evaluation of operational weather forecast models by several specially equipped supersites (Illingworth et al., 2007). Every supersite operates a cloud radar, a LiDAR, a microwave radiometer and a rain gauge. The Cloudnet algorithms combine all measurements and information about the temperature, humidity and wind speed profile from a model to a synergetic Target Classification. The valuable classification provides detailed information about the cloud structure, phase, composition and development of clouds. Nevertheless, this product is so far only used as a basis to derive model variables like liquid water content (LWC) and other parameters from the measurements.

Daily and monthly statistics of the cloud fraction, LWC and ice water content (IWC) are generated operationally for the observations and various weather forecast models to monitor forecast quality. Those Cloudnet quantities serve as a reference for various cloud evaluation studies of atmospheric models like Hogan et al. (2009); Barrett et al. (2009); Bouniol et al. (2010) and Illingworth et al. (2015b). Further studies used the derived LWC, IWC and cloud fraction e.g. to calculate cloud radiative effects (Ebell et al., 2011), to investigate different cloud parameterization schemes (Morcrette et al., 2012) or to compare measurements of CloudSat and CALIPSO (Cloud-Aerosol Lidar and Infrared Pathfinder Satellite Observation) with ground-based observations (Protat et al., 2010). The Cloudnet measurements and products are also utilised e.g. to investigate mixed-phase clouds in Central Europe (Bühl et al., 2016). Nevertheless, the Cloudnet Target Classification has yet not been directly used for the evaluation of cloud's representation within a model, because there is no appropriate model output to compare with. A modelled classification surrogate offer a great potential for model evaluation studies or e.g. investigations of mixed-phase clouds of atmospheric models.

Furthermore, similar cloud classifications are available from space-borne instruments and processed by for example the DARDAR algorithms (Delanoë and Hogan, 2010) which shows the huge potential for a consistent model-based surrogate for detailed cloud evaluation. The International Satellite Cloud Climatology Project (ISCCP) (Schiffer and Rossow, 1983; Rossow and Schiffer, 1991) and studies like the ones from Derrien and Gléau (2005) and Parikh (1977) investigate two-dimensional cloud classifications based on satellite data. Another approach is to use rain radars for a hydrometeor classification (Straka et al., 2000; Liu and Chandrasekar, 2000; Lim et al., 2005; Park et al., 2009) providing detailed cloud information during precipitation.

This study presents a simple cloud classification algorithm for numerical atmospheric models to generate a consistent product to the Cloudnet Target Classification for further analysis (Fig. 1). Standard error metrics like BIAS, RMSE or skill scores aren't appropriate for a cloud classification comparison due to the multi-dimensional and categorical problem as well as due to the large variability of clouds in time and space. Similar challenges exist at the validation of precipitation forecasts (Ebert, 2008). Fuzzy verification methods incorporate the large variability and small scales on which clouds and precipitation act and thus allow forecasts to be uncertain in time and space, which are commonly used for the evaluation of precipitation (Zimmer and Wernli, 2011; Ebert, 2009; Amodei and Stein, 2009; Weusthoff et al., 2009; Ament et al., 2011). These techniques are as well used for satellite-based cloud classifications (Jin, 2016).

The novel cloud classification algorithm is presented (section 2) and several example evaluation techniques utilising this new product are shown (section 3). The mean cloud statistics separated by different cloud categories are analysed by the

frequency of occurrence. The accuracy of cloud forecasts for the right time and place is investigated by direct comparisons of the modelled and measured cloud classification for which also fuzzy verification methods are applied. The potential of this new evaluation approach is shown by comparisons with the operational COnsortium for Small-scale MOdelling (COSMO) model for the German domain (COSMO-DE) as well as with first ICOsahedral Non-hydrostatic (ICON) Large Eddy Simulations (LES) (section 4). Results are finally concluded and discussed (section 5).

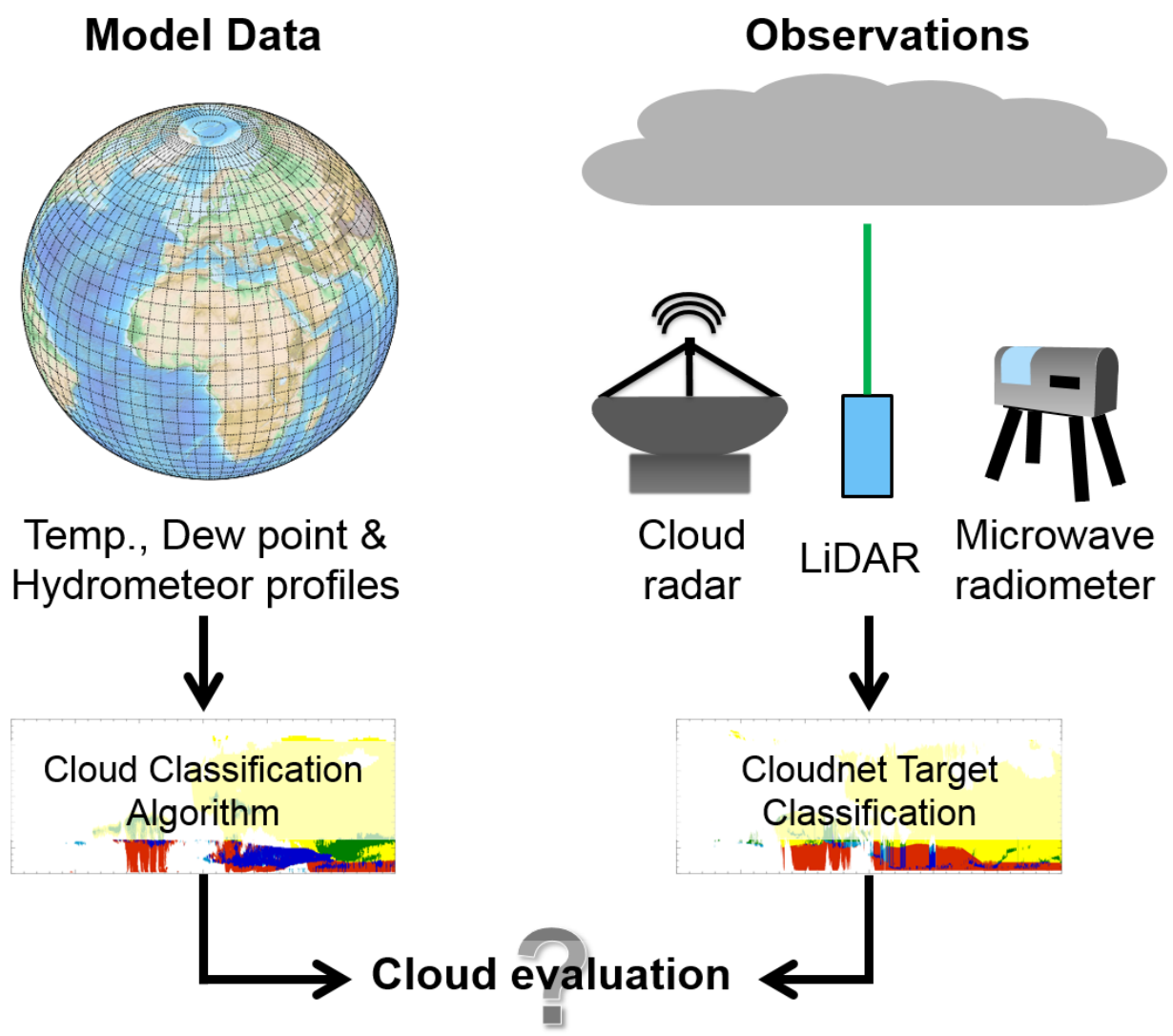

**Figure 1.** Schematic illustration of the cloud classification approach to generate a comparable Cloudnet target classification on the basis of an atmospheric model output. The original target classification is obtained by observations, using cloud radar, LiDAR and microwave radiometer measurements.

## 2 Data, Methods and Cloud Classification Algorithm

The observations and derived Cloudnet products are obtained by the Leipzig Aerosol and Cloud Remote Observations System (LACROS) supersite of the HOPE campaign (Macke et al., 2017) for April and May 2013 (SAMD, 2018). This supersite was located at a sewage plant near Krauthausen, Germany (40 km west of Cologne). Various cloud types like low-level cumulus clouds, high cirrus clouds, large precipitating ice clouds and several frontal passages were captured during these two months depicting a large variety of typical synoptic situations. The Cloudnet products have a time resolution of 30 seconds and a height resolution of 30 m. The data is available from roughly 200 m above ground up to 15 km height due to the remote sensing measurement characteristics.

The operational COSMO-DE model of the German Meteorological Service (DWD) is evaluated (Baldauf et al., 2011). This cloud-resolving model runs at a horizontal resolution of 2.8 km and contains 51 height levels with terrain following hybrid coordinates. The layer thickness is increasing with height till the upper edge of the model at 22 km. The 1-hourly output

of 12 h forecasts, starting at 00 and 12 UTC, are analysed within this study. The COSMO-DE model gets its initial and hourly boundary conditions from the COSMO model setup with 7 km grid spacing covering Central Europe (COSMO-EU). Microphysical processes are parameterised by a 1-moment bulk formulation (Baldauf et al., 2011), which contains five different hydrometeor classes (specific cloud water content *QC*, specific cloud ice content *QI*, specific rain water content *QR*, specific snow content *QS* and specific graupel content *QG*). The closest grid point to the LACROS supersite is selected for the point-to-point comparisons and a region of roughly 50 x 50 km² between Aachen and Düsseldorf is extracted from the model output for the fuzzy verification.

The new cloud classification algorithm combines the temperature, dew point and all hydrometeor information of an atmospheric model to generate a consistent cloud classification corresponding to the Cloudnet Target Classification. The cloud classes are determined based on physical principles from the model output by consecutive case selections, as depicted in Fig. 2. Every grid box of the model is assessed independently by the algorithm.

The official Cloudnet algorithm differentiates between eleven categories, which are "Aerosol & insects", "Insects", "Aerosol", "Melting ice & cloud droplets", "Melting ice", "Ice & supercooled droplets", "Ice", "Drizzle/rain & cloud droplets", "Drizzle or rain", "Cloud droplets only" and "Clear sky". The aerosol, as well as insect categories, are not diagnosed by the presented cloud classification algorithm, because most atmospheric models like COSMO-DE don't provide any information about it. Therefore, these categories are set to "Clear sky" in the observations and simulations. The remaining eight cloud classes are defined consistently with respect to the Cloudnet algorithms where possible. For example, the temperature has to be above the freezing level for liquid phases as for example "Drizzle or rain" and the dew point below 273.15 K for ice classes like "Ice & supercooled droplets". The snow and graupel hydrometeors are also classified as ice because of their ice phase and characteristics. Other distinctions between e.g. "Drizzle or rain" and "Drizzle/rain & cloud droplets" are set by the specific cloud water hydrometeor concentration *QC* for the cloud classification algorithm. In contrast, the Cloudnet algorithm is based on remote-sensing observations and thus using e.g. thresholds of the cloud radar reflectivity and LiDAR attenuated backscatter coefficient to differentiate between both classes. Nevertheless, a comparable diagnosis is developed to generate a consistent synthetic cloud classification for the model with respect to Cloudnet.

The algorithm itself works on every grid box as follows. The category of "Ice" is e.g. determined by a dew point below the freezing point and a dominant concentration of cloud ice defined by an ice concentration *QI* larger than a certain threshold and a cloud water concentration *QC* below another fixed threshold. The "Melting ice" category is e.g. defined by a dew point greater than the freezing point and *QI* greater than the critical hydrometeor concentration. All rules are compiled together by the flowchart in figure 2. The order of the case selection statements is crucial to get physical consistent results. If for example a grid box is first checked for "Drizzle/rain & cloud droplets" with a temperature above the freezing level, *QR* and *QC* larger than a certain threshold and afterwards examined for "Drizzle or rain", for which only the first two conditions have to be fulfilled, all "Drizzle/rain & cloud droplets" cases would be classified only as "Drizzle or rain". Similar situations exist for other case selection statements like "Melting ice" and "Melting ice & cloud droplets". The algorithm can be easily advanced by additional case differentiations including for example information about the surrounding grid boxes or aerosols & dust, if they are provided by the model.

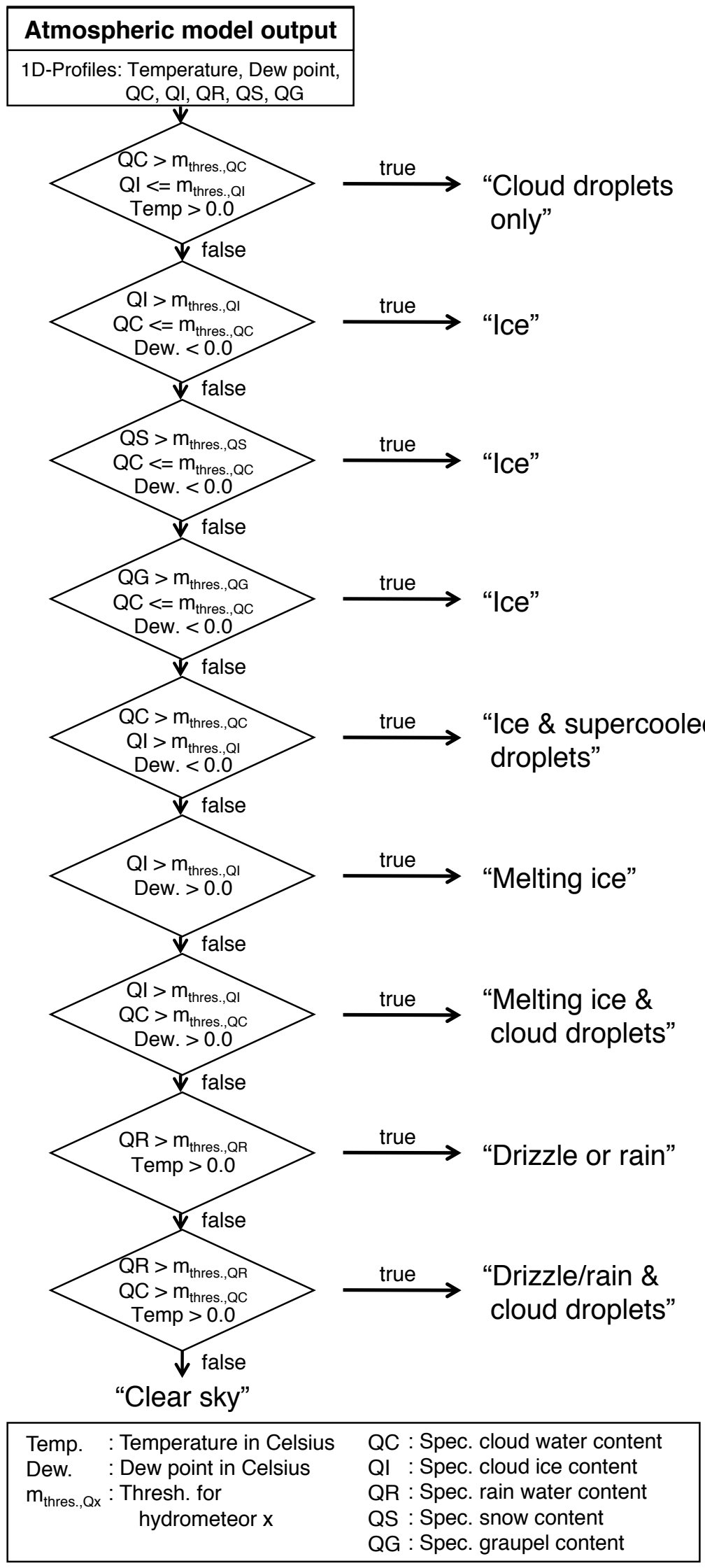

**Figure 2.** Flowchart of the cloud classification algorithm using the output of an atmospheric model to generate a comparable Cloudnet target classification, differentiating eight different cloud categories.

The threshold values for the different hydrometeor concentrations have to be chosen in accordance with the characteristics of the Cloudnet instruments and the atmospheric model for a physically consistent comparison. The instruments' sensitivity is influenced by several parameters like the technical configuration as well as certain atmospheric conditions. For that reason, physically determined threshold values are hard to define and would imply various assumptions. The results would be thus complicated to interpret. From the model perspective, a significant hydrometeor concentration of the COSMO-DE and ICON model is roughly 1e-7 kg/kg (A. Seifert, personal communication, February 9, 2018). Therefore, this threshold is used

for all hydrometeor classes at the cloud classification algorithm in this study. The simplification of using only one model-based threshold has nevertheless to be considered when interpreting the results but reduces at the same time the number of assumptions.

One big advantage of the algorithm is its simplicity, which allows for a fast application on new model data. In addition, all case selections are clear and easily comprehensible. Despite that the Cloudnet Target classification is based on measurement data, the introduced cloud classification algorithm generates a consistent surrogate for atmospheric models. This enables physical consistent comparisons using the valuable cloud classification tool.

The high-resolution Cloudnet observations are coarse-grained to the lower temporal and spatial resolution of the COSMO-DE by averaging the categories to the most frequent one for each grid box individually. The variability of the observed cloud classes at the coarse-graining is rather low due to the already cloud-resolving resolution of the COSMO-DE and the scale of clouds instead of small aerosols and dust. The cloud classification is evaluated by the mean frequency of occurrence of the different cloud categories to analyse the statistics. The cloud forecast quality is investigated by the hit rates between the observed and modelled classification. A hit is defined as the match of the same category at the same time and location thus requiring an exact match of the simulation with the observation. Due to possible lags of a forecast or uncertainties in time and space of the model, fuzzy verification methods are tested. The fuzzy verification allows predictions to be uncertain in time, in space or in both dimensions. If for example the observation contains "Ice" at 3 km height at 12 UTC and the model doesn't contain "Ice" at that time at the closest grid point but at a neighbouring one, it's only uncertain in space and thus rated as a hit by the fuzzy verification. The same case is also applicable for being uncertain in time if e.g. the model contains an "Ice" box at 13 UTC instead of 12 UTC. A bootstrapping with return (Efron and Tibshirani, 1993) is furthermore applied to the cloud classification to give an uncertainty estimation of the results shown within this study.

## 3 Exploring the Cloud Classification method for model evaluation

The novel cloud classification algorithm generates a consistent product with respect to the observed Cloudnet Target classification for atmospheric models. The two datasets can thus be compared directly and the same statistical analysis can be applied to the model and the observations. The cloud classification provides an easy but at the same time comprehensive view of the cloud structure, phase and development which indicates also the overall model performance. The qualitative comparison also shows indirectly the quality of the temperature-/humidity profile as well as of the cloud microphysics.

In addition to the general cloud fraction analysis of the Cloudnet project, the new cloud classification of the model can be used to evaluate the vertical cloud distribution regarding different cloud phases. The frequency of occurrence for the multiple cloud classes presents detailed insights into the cloud statistics and the underlying cloud microphysics, which helps to find possible shortcomings of current models (see section 3.1). The direct comparison of both classifications can also be used to investigate the accuracy of cloud forecasts differentiated by the cloud categories from a weather forecast perspective. Therefore, weather predictions can be analysed, if the right cloud type is simulated at the right time and place (see section 3.2), which is e.g. important for the different radiative properties of liquid and ice clouds. Nevertheless, due to the high temporal and spatial

variability, especially of small cumulus clouds, an exact match of the simulation and the observation isn't expected. For this reason, the two datasets are examined by using fuzzy verification methods, allowing the model to be uncertain in time and space to consider a simple time lag or displacement of cloud forecasts (see section 3.3).

The mentioned evaluation methods using the cloud classification are exemplarily tested by the operational COSMO-DE analyses respectively forecasts of April and May 2013. The cloud classification algorithm is applied to this model dataset and compared with the corresponding coarse-grained LACROS Cloudnet Target Classification (Fig. 3).

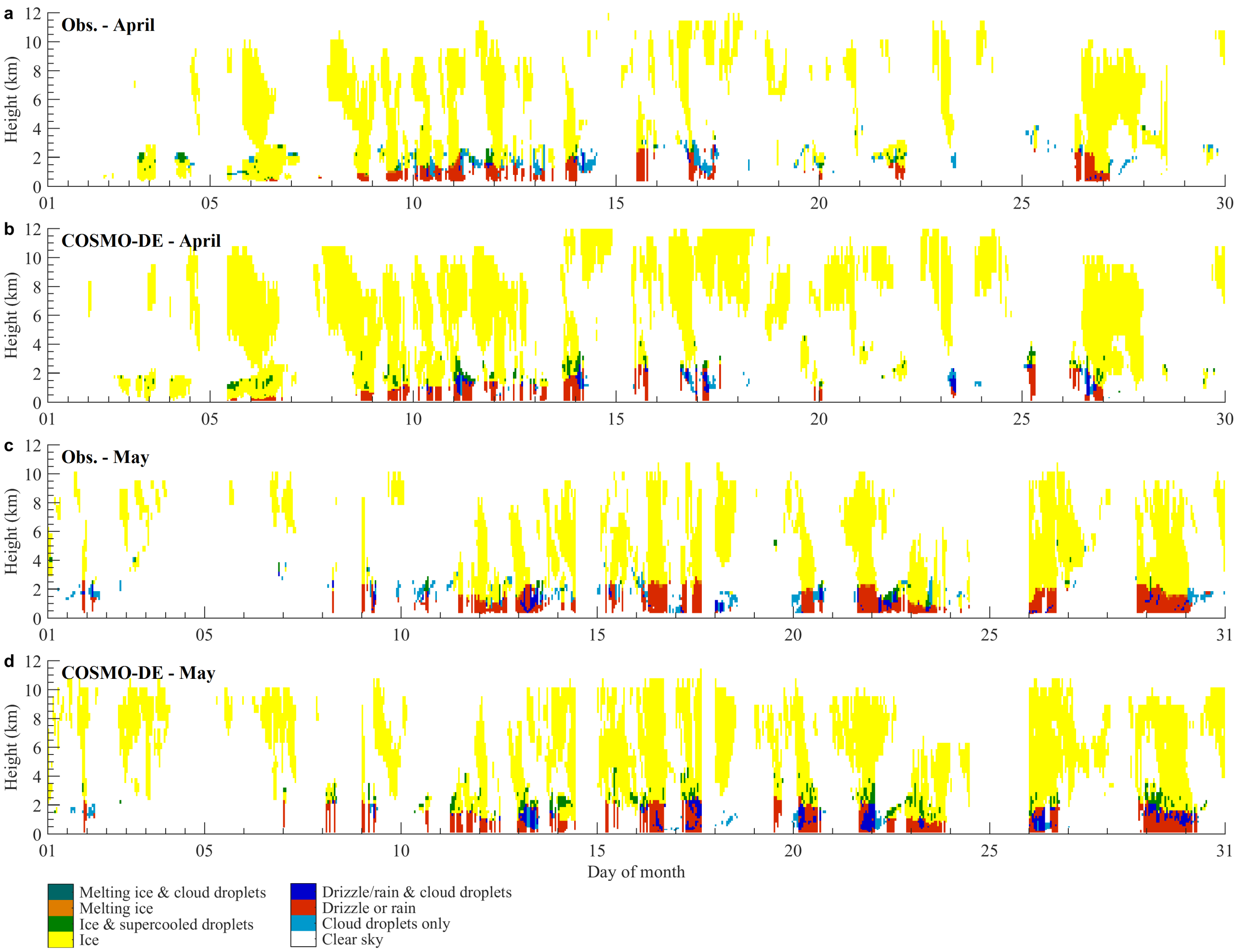

**Figure 3.** Cloudnet Target Classification of the LACROS observations (a,c) and of the cloud classification algorithm applied on the COSMO-DE (b,d) for April (a,b) and May (c,d) 2013. The output is provided on the 51 height levels of the COSMO-DE with an hourly resolution. The observations are adapted to the common grid by using the most frequent category.

At first glance, the qualitative comparison of the new synthetic cloud classification of the COSMO-DE shows similar cloud patterns and types. A proper design of the algorithm is indicated by the occurrence of all observed categories at the modelled

classification. In detail, more "Ice" is found at the model above 3 km compared to the observations during both months. Other differences are visible, but overall both classifications are in good agreement. The general cloud structure, as well as the time of occurrence of rain events, are very similar, see for example May.

### 3.1 Frequency of Occurrence of Cloud Classification

The frequency of occurrences of the cloud classes are calculated for the observations and model to quantify differences between both datasets and investigate qualitative findings in more detail. Therefore, the cloud statistics of the model can be evaluated considering the multiple cloud phases of the classification. The mean vertical cloud profile for the cloud classes highlights e.g. certain biases or model shortcomings for specific cloud types like ice clouds and thus allow for in-depth analysis of the cloud microphysics. The cloud type climatology of the model for specific locations can be assessed using a long time series. To illustrate the potential of this method, the frequency of occurrence is calculated for the COSMO-DE dataset. The eight cloud categories are merged to four because of the small number of occurrence of most categories as well as for reasons of clarity and comprehensibility. In respect to the model hydrometeors of cloud water *QC*, cloud ice *QI*, snow *QS*, graupel *QG* and rain *QR*, the cloud classification categories are merged to "Clear Sky", "Ice Clouds", "Liquid Clouds" and "Drizzle or rain". For this, the categories of "Drizzle/rain & cloud droplets" and "Cloud droplets" are merged to a new category named "Liquid clouds". The categories of "Ice", "Ice & supercooled droplets", "Melting ice" and "Melting ice & cloud droplets" are combined to "Ice clouds". The previously mentioned classes are also merged, because "Ice Clouds" mainly consists of "Ice" and "Liquid clouds" out of "Cloud droplets only". The categories of "Clear sky" and "Drizzle or rain" aren't modified. The timely averaged frequencies of occurrence profiles for all four cloud categories are depicted in figure 4.

The analysis points out model biases, seen for example by a continuous overestimation of "Ice clouds" by up to 30 % above 3 km and "Liquid clouds" by up to 3 % between 1 and 3 km, which would not be feasible to determine looking only at the cloud fraction statistics. Therefore, the "Ice cloud" overestimation explains the differences found at the "Clear sky" category with an underestimation of a similar magnitude with roughly 30 %. This also confirms the previously stated results of the qualitative comparison. Nevertheless, especially at high altitudes, the lower sensitivity of the remote-sensing instruments has to be considered which could reduce the observed frequency of occurrence of "Ice clouds". The choice of merging the categories for "Ice clouds" doesn't affect the overall conclusions because of the small number of other categories like for "Drizzle/rain & cloud droplets" with only 251 occurrences out of 47,576 samples.

The profile of "Drizzle or rain" fits well to the observations which was also seen at the qualitative comparison. Precipitation or at least Drizzle was present for roughly 20 % of the time of the two months. The missing "Drizzle or rain" within the lowest layers at the observations are due to the remote-sensing instruments, which start measuring roughly 200 m above ground. COSMO-DE captures this category with a frequency of occurrence of 20 % down to the ground. Overall, the generally good statistics of the COSMO-DE except for the mismatches found at "Ice clouds" is proven by the similar shape of the distributions of the four distinct categories.

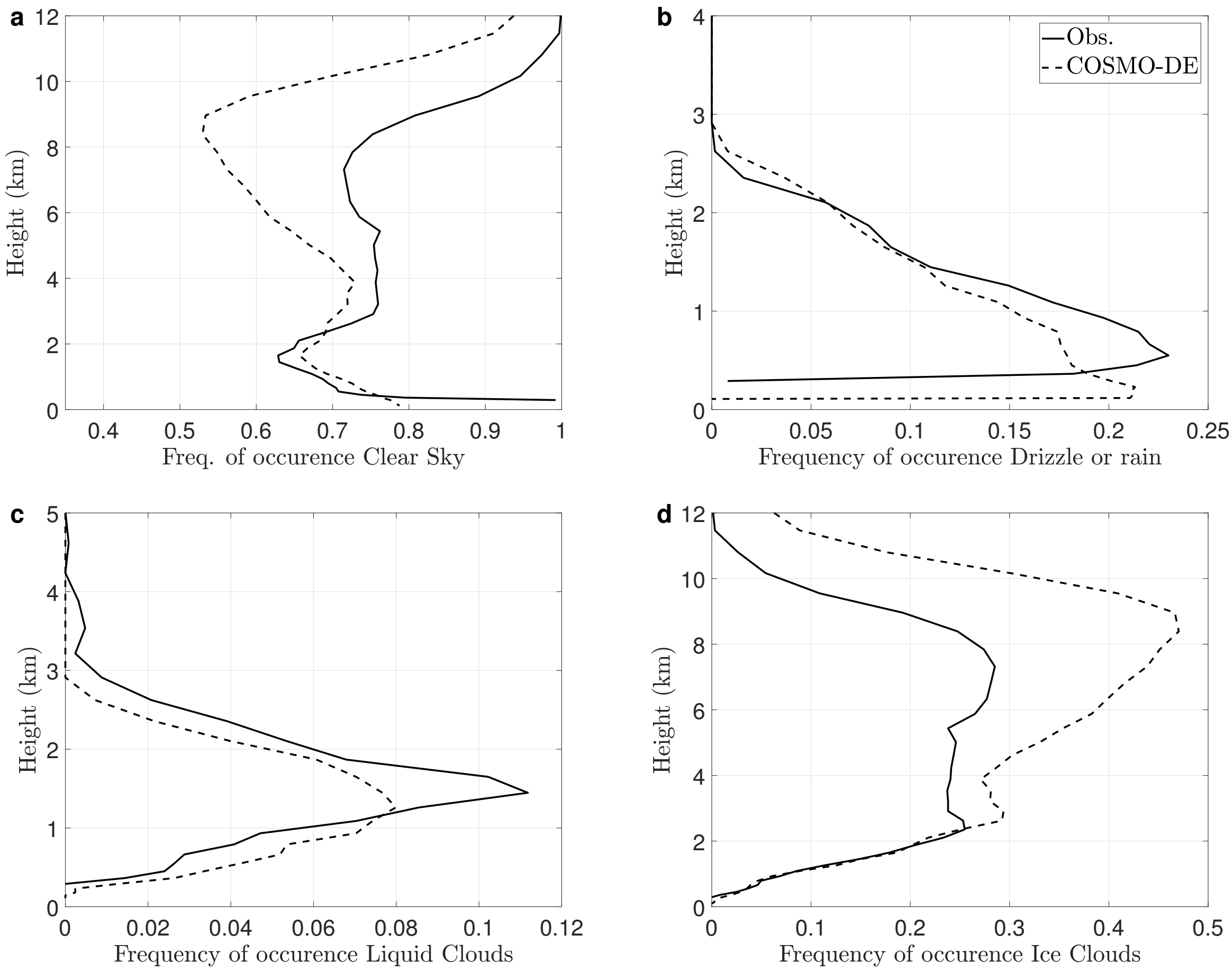

**Figure 4.** Frequency of occurrence (0-12 km) of April and May 2013 of the Clear Sky (a), Drizzle or Rain (b), Liquid Clouds (c) and Ice Clouds (d) category for the LACROS Cloudnet target classification (black solid line) and for the COSMO-DE Cloud classification (black dashed line). Mind different axis.

### 3.2 Point-to-Point Verification

Precise cloud predictions for the right time and place are of high importance for various applications like general weather forecasts or even for the radiative budget and all related quantities of the model itself. The cloud classification contains for every height interval of each time step information about the cloud phase, which can be directly compared between the observed and the modelled classification. This enables to analyse if the model predicts the correct cloud type at the same time and location as of the measurements even though an exact matching is not expected due to the chaotic characteristics of the atmosphere. The evaluation of the cloud composition is e.g. crucial to investigate their radiative properties and derived quantities. Certain mismatches of the multiple cloud classes can furthermore indicate model shortcomings. For example, "Cloud droplets" of the

**Table 1.** Point wise comparison (Contingency table) of the eight different cloud categories of the LACROS Cloudnet target classification and of the COSMO-DE Cloud classifications of April and May 2013. The COSMO Cloud classification results are based on the 00/12 UTC analyses with the hourly forecasts for the hours in between. The absolute numbers of the contingency table are normalized by the total number of observed events of each category.

| | | | **LACROS Observations** | | | | | | | |
|---|---|---|---|---|---|---|---|---|---|---|
| | | Overall number | Clear sky | Cloud droplets | Drizzle or rain | Drizzle /rain & cloud droplets | Ice | Ice & supercooled droplets | Melting ice | Melting ice & cloud droplets |
| **COSMO-DE** | Clear Sky | 37137 | 84.6 | 55.3 | 31.6 | 25.9 | 18.6 | 34.0 | 6.6 | 0.0 |
| | Cloud droplets | 691 | 0.2 | 8.7 | 4.2 | 12.4 | 0.0 | 0.0 | 0.0 | 0.0 |
| | Drizzle or rain | 2432 | 2.4 | 8.5 | 50.3 | 31.1 | 1.0 | 3.0 | 52.5 | 0.0 |
| | Drizzle/rain & cloud drop. | 251 | 0.3 | 8.4 | 13.2 | 27.5 | 0.5 | 2.5 | 26.2 | 0.0 |
| | Ice | 6804 | 12.0 | 13.6 | 0.5 | 2.4 | 75.3 | 43.5 | 14.8 | 0.0 |
| | Ice & supercooled droplets | 200 | 0.4 | 5.5 | 0.2 | 0.8 | 4.6 | 17.0 | 0.0 | 0.0 |
| | Melting ice | 61 | 0.0 | 0.0 | 0.0 | 0.0 | 0.0 | 0.0 | 0.0 | 0.0 |
| | Melting ice & cloud drop. | 0 | 0.0 | 0.0 | 0.0 | 0.0 | 0.0 | 0.0 | 0.0 | 0.0 |

modelled classification, which are classified as "Drizzle or rain" by the observations suggest e.g. problems at the rain formation process.

As an example, the COSMO-DE Cloud Classification is compared pointwise with the LACROS observations for which the results are discussed. In total 47,576 points with 8 different cloud categories are analysed, which results in an 8 x 8 contingency table (Tab. 1). The table depicts how often the model and the observations contain the same category at the same time, place and height, respectively how often another category is predicted than measured. Therefore, hits on the diagonal are the optimum where the model matches the observations.

The high number of "Clear sky" cases and the simplicity to diagnose this category by the algorithm results in the best agreement of 84.6 % found for "Clear sky". The overestimation of "Ice" above 3 km is depicted by a hit rate of only 75.3 %. In addition, 12 % of the modelled "Ice" points are thus observed as "Clear sky".

General issues of atmospheric models like the right representation of the melting layer are identifiable by this analysis. For example, a lower melting layer through a warmer temperature profile lead to an earlier melting of ice hydrometeors, which is seen by the observed 52.5 % of "Melting ice" points already modelled as "Drizzle or rain". Difficulties at the right distinction of the eight different cloud categories are visible e.g. by 31 % of observed "Drizzle/rain & cloud droplet" points, which are categorised as "Drizzle or rain" at the modelled classification. The underestimation of "Liquid clouds" (sect. 3.1) is confirmed by 55.3 % of the modelled "Clear sky" points measured as "Cloud droplets". Only a few observations were made of rare categories like "Drizzle/rain & cloud droplets" with 251 points or even "Ice & supercooled droplets" with 200 points out of

47,576. Therefore, the right capturing of those categories is very difficult for the model because of the small sample size and thus only low hit rates are obtained like 17 % for "Ice & supercooled droplets".

### 3.3 Fuzzy Verification

Small displacements or time lags of cloud and precipitation forecasts often induce large errors, if the model output is compared pointwise with the observations. Depending on the specific interest as for example to evaluate the statistics of the cloud forecasts, fuzzy verification methods are more appropriate allowing the model to be uncertain in time and/or space or further dimensions like the cloud phase. In addition, the large variability of clouds, the multi-dimensional and categorical problem of the classification as well as different resolutions of the observations and the model, cause problems at using standard point to point verification metrics like BIAS and RMSE. Therefore, fuzzy verification techniques are more appropriate for the analysis of the cloud classification to assess a small shift in time or place of a cloud still as a correct prediction.

For each of the fuzzy analyses like e.g. being fuzzy in space by one grid box, an 8 x 8 contingency table can be calculated, which would be difficult to interpret due to the large amount of numbers. For that reason, the focus at this point is on the hit rates of the same cloud phase between the model and the measurement to evaluate the overall accuracy of the cloud forecasts including the cloud type. Further comparisons with a random forecast or a bootstrapping can be applied to the cloud classification to investigate the statistical significance of the results. The fuzzy verification of the cloud classification is exemplarily tested by the two months of COSMO-DE and LACROS observation dataset.

For the fuzzy verification in time, the hour before and after the observation is included, shown by the "Time" column in Tab. 2. For being fuzzy in space, first one grid box around the centre ("Space 1" column) and then three ("Space 3" column) grid boxes surrounding the middle as well as the whole extracted area ("Space Full" column) of 18 x 17 grid points (50 x 48 km) are considered for the evaluation. Only the hit rates, normalised by the total number of observed events for each category separately, are regarded. As a benchmark, we calculated 10,000 times a hit rate resulting from randomly chosen observational and model grid points and computed the average. The statistical significance of the point to point comparison is tested by a bootstrapping with return applied to the cloud classification with 10,000 iterations. The bootstrapping assumes that the climatology is captured correctly, which is indicated by a good agreement of the frequency of occurrences seen in section 3.1. The standard deviation of the bootstrapping provides an uncertainty estimation of the analysis. All results are compiled in table 2.

The easier prediction of classes like "Clear sky" than of more rare categories is obvious by a hit rate of roughly 72.7 % for "Clear sky" already gained by the comparison with a random forecast. Therefore, the added value for "Clear sky" isn't that high as indicated by the large hit rate of the real forecast of 84.6 %. The improved forecast skill is especially visible for rare categories like "Drizzle or rain", where an increase of 44 % and a factor of eight is found compared to the random forecast. The reliability of the results, based on the bootstrapping, is higher for more frequent occurring categories like "Clear sky" with a small standard deviation of less than 0.2 % compared to 2.3 % for "Ice & supercooled droplets" with only 200 observed points. Overall, standard deviations of less than 2.3 % prove the statistical significance of the presented results.

**Table 2.** Hit rate table of the eight different cloud categories of the observed LACROS Cloudnet target classification and of the COSMO Cloud Classification of April and May 2013. The hit rates are normalized by the total number of observed events (first column) of each category. The point to point comparison results are at the first numbers of the first rows of the second column. The numbers at the second rows of the second column are the mean hit rates of randomly chosen points from 10,000 iterations. The second numbers of the first rows at the second column are the standard deviation calculated by a bootstrapping with 10,000 iterations. The hit rates of the third column are determined by being fuzzy in time by one hour. The hit rates of being fuzzy in space are at the fourth to sixth column. For more details, see text.

| | | **LACROS Observations** | | | | | |
|---|---|---|---|---|---|---|---|
| | | Overall number | Point/Point Hits Rel | Time Hits Rel | Space 1 Hits Rel | Space 3 Hits Rel | Space Full Hits Rel |
| **COSMO-DE** | Clear Sky | 37137 | 84.6 ± 0.2<br>72.7 | 91.8 | 87.4 | 91.2 | 95.4 |
| | Cloud droplets | 691 | 8.7 ± 1.3<br>0.5 | 17.9 | 15.1 | 24.0 | 40.5 |
| | Drizzle or rain | 2432 | 50.3 ± 1.0<br>6.3 | 71.5 | 62.8 | 74.5 | 85.7 |
| | Drizzle/rain & cloud drop. | 251 | 27.5 ± 1.4<br>1.2 | 40.6 | 35.1 | 45.8 | 60.2 |
| | Ice | 6804 | 75.3 ± 0.5<br>18.2 | 88.7 | 82.5 | 88.8 | 94.1 |
| | Ice & supercooled droplets | 200 | 17.0 ± 2.3<br>1.0 | 26.0 | 28.5 | 44.0 | 62.5 |
| | Melting ice | 61 | 0.0 ± 0.0<br>0.0 | 0.0 | 0.0 | 0.0 | 0.0 |
| | Melting ice & cloud drop. | 0 | 0.0 ± 0.0<br>0.0 | 0.0 | 0.0 | 0.0 | 0.0 |

As expected, by being fuzzy in time or space, the hit rates increase by considering more and more grid boxes respectively time steps. Thus, including the whole region with 18 x 17 grid points ("Space Full" column) shows the highest hit rates. Larger variability within three hours of the model ("Time" column) is observable compared to 9 considered grid boxes of being fuzzy by one additional grid box ("Space 1" column), seen by higher hit rates for being fuzzy in time than in space, except for "Ice & supercooled droplets". The opposite is found regarding three or more surrounding grid boxes, except for "Clear Sky". The biggest rise at the hit rates for the fuzzy verification is seen for rare categories like "Cloud droplets". For this category, an increase by a factor of four from 8.7 % to 40.5 % is visible comparing the point-to-point verification with the fuzzy verification including all extracted points. The presented results indicate for possible model improvements and are a good starting point for further in-depth analysis.

## 4 Large Eddy Simulation Cloud Classification

The applicability of the presented cloud classification algorithm to other atmospheric models is tested by a case study with the ICON LES model (Dipankar et al., 2015). Realistic Germany-wide LES simulations were performed within the High Definition Clouds and Precipitation for Advancing Climate Prediction (HD(CP)²) project with a horizontal resolution down to 156 m and an output frequency of 9 seconds for specific locations like the Cloudnet supersites (Heinze et al., 2017). Thus, the output is averaged to the 30 sec. resolution of Cloudnet using the most frequent cloud class for each time interval. The ICON LES model consists of 151 terrain following levels up to 22 km with increasing layer thickness with altitude. The initial-/boundary conditions are provided by the COSMO-DE analysis and boundary conditions are updated every hour. The simulations were done with the two-moment microphysics of (Seifert and Beheng, 2001), which has 6 hydrometeor categories (cloud water, cloud ice, rain water, snow, hail and graupel). According to Jerger (2014), hail could be assigned to ice as well, which allows applying the same cloud classification algorithm (Fig. 2) to the LES output as for the COSMO-DE simulations. The algorithm is applied for a case study to the nearest grid box of the LACROS supersite for 26 April 2013, showing a frontal passage during noon (Fig. 5).

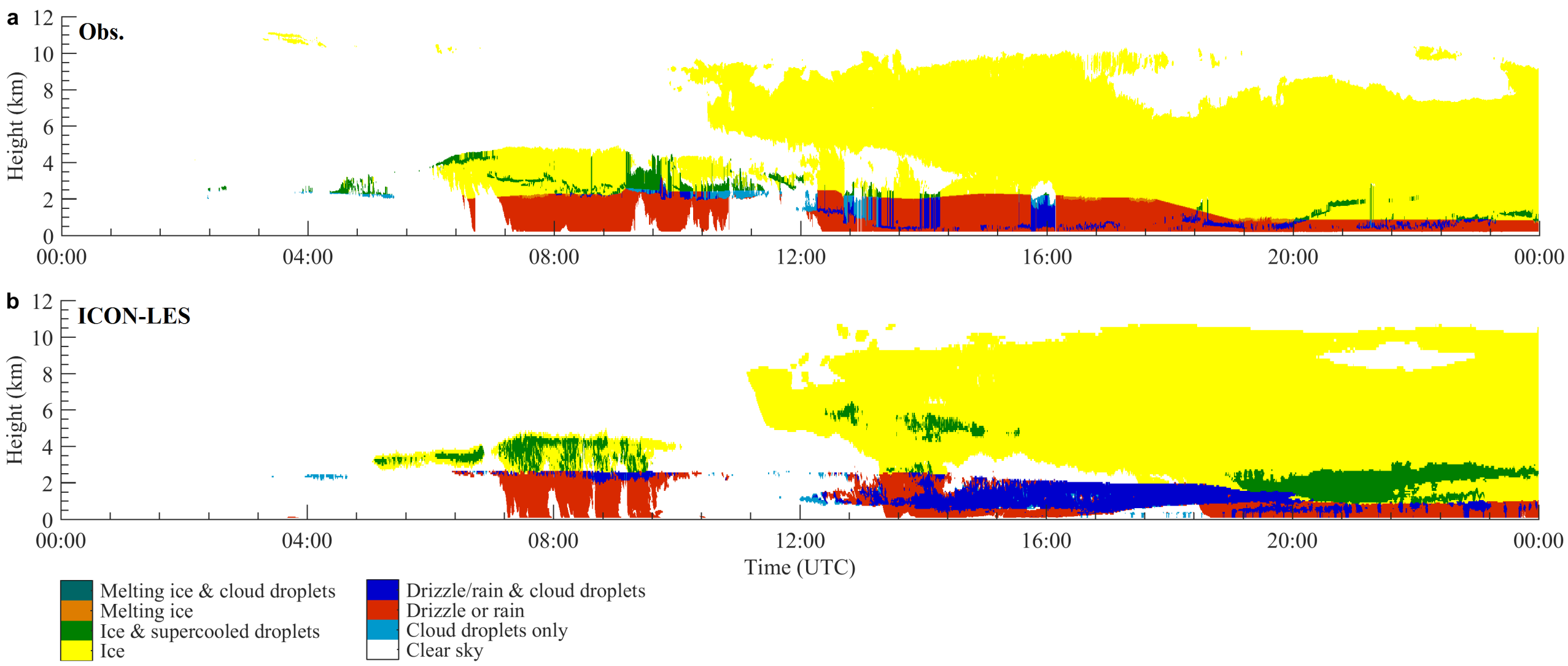

**Figure 5.** Cloud classification for 26 April 2013 using Cloudnet LACROS measurement data (a) and ICON LES Cloud Classification (b).

In general, a similar cloud structure and phase is found comparing the modelled cloud classification to the observed one. Nevertheless, differences for example between the categories of "Drizzle or rain" and "Drizzle/rain & cloud droplets" can be identified, which provide detailed insights into the models' microphysics. The example shows the usability of the developed cloud classification algorithm as well for other models even at LES resolution. Also, the other introduced cloud evaluation methods can thus be applied, which is shown exemplarily by the frequency of occurrence for the four cloud classes (Fig. 6).

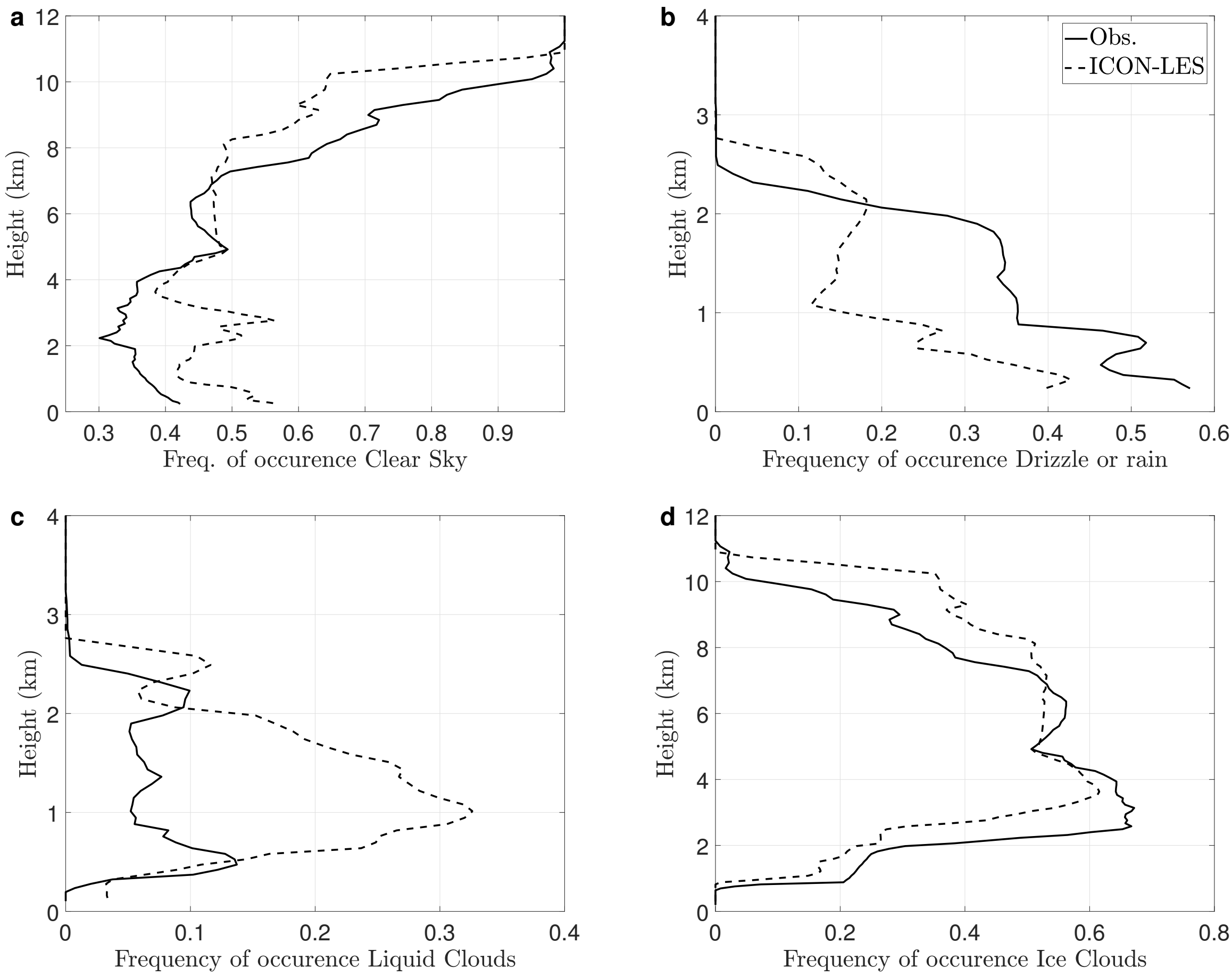

**Figure 6.** Frequency of occurrence (0-12 km) of 26 April 2013 of the Clear Sky (a), Drizzle or Rain (b), Liquid Clouds (c) and Ice Clouds (d) category for the LACROS Cloudnet target classification (black solid line) and for the ICON-LES cloud classification (black dashed line). Mind different axis.

Rain events of the ICON model seem to be too rare and underestimated by roughly 15 %, as also found by Heinze et al. (2017). This results in an overestimation of the "Clear Sky" frequency of occurrence. The profile of "Ice clouds" is overall in good agreement with the observations. Nevertheless, an overestimation of "Ice clouds" by roughly 10-20 % above 7 km is found which might arise from the lower sensitivity of the remote sensing instruments and thus less observed ice at high altitudes. The low "Liquid cloud" layer during the afternoon, seen as "Drizzle or rain" by the observations, leads to an overestimation of this category by the model, indicating issues at the rain formation process.

Nevertheless, the high spatial and temporal resolution of the ICON output shows an unprecedented detail of the modelled cloud structure and development, which shows among others the added value of these realistic large eddy simulations. Addi-

tionally, the cloud classification offers a very intuitive and comprehensive view on the modelled clouds to make big datasets of large LES accessible, which is crucial e.g. to find model issues or screen the simulations for interesting physical processes.

## 5 Conclusions and Discussion

The proposed cloud classification algorithm uses the temperature, dew point and hydrometeor profiles of a numerical atmospheric model to generate a cloud classification similar to the observational-based Cloudnet Target Classification. This observational product is a comprehensive tool for the detailed evaluation of cloud phase, composition and structure, but has so far only been used to derive model quantities like cloud fraction or ice water content from the observations. The modelled surrogate makes therefore a direct comparison of the extensive cloud classification product possible, allowing for an in-depth cloud evaluation. The modelled cloud classification provides, in addition, an easily accessible first impression of the vertical cloud structure. The qualitative comparison with the observations indicates how well the clouds are represented by the model, giving a hint about the overall model performance and the underlying cloud microphysics.

The comparably designed cloud classification can consequently serve ideal as a basis for various statistical analyses and further derived cloud properties. For example, the frequency of occurrence can be calculated to evaluate the mean vertical cloud distributions for the multiple cloud types. Model biases of certain cloud types or other shortcomings of the model can thus be identified. Furthermore, the prediction of the right cloud type at the same time and location can be investigated by comparing the cloud phase of every height level for each time step with the observations. Fuzzy verification techniques are more appropriate to assess cloud forecast statistics due to the different time and spatial resolutions of the observations and the model as well as due to the large variability of clouds. This is especially worthwhile because of the multi-dimensional and categorical dataset of the cloud classifications. The fuzzy verification allows the model to be uncertain in time and space, which prevents a misinterpretation of the evaluation due to e.g. a simple time lag or displacement of the model.

The cloud classification algorithm and evaluation methods are exemplarily tested by comparing the COSMO-DE model data with two months, respectively the ICON LES model with one day, of observations for a mid-latitude Cloudnet supersite. The results show the value of the classification to point out, for example, certain model shortcomings. The calculated frequency of occurrence of "Ice clouds" shows a significant overestimation above 3 km with up to 30 % for the COSMO-DE. Thus, a too low frequency of occurrence for "Clear sky" conditions by up to 30 % is seen in the middle and upper atmosphere. The pointwise comparison of the cloud classification demonstrates e.g. modelled "Drizzle or rain" cases, which are observed as "Melting ice" points. The earlier phase transition to rain indicates possible issues of a too warm temperature profile or of the cloud microphysics. Allowing the model to be uncertain in time or space leads, as expected, to higher hit rates. The hit rates of the correct cloud category are higher for being uncertain by one hour than of being uncertain by one or more grid boxes surrounding the centre for the COSMO-DE. Considering three or more cells, the opposite is the case. The application of the new cloud classification algorithm to the ICON LES provides very detailed information about the cloud structure and phase at a similar resolution as of the observations.

The shown results indicate a plausible design of the cloud classification algorithm as well as for reasonably chosen threshold values. Nevertheless, an approach of a neural network or machine learning could be worthwhile to test to find threshold values or include more parameters to the calculation of the classification like Bankert (1994) and Azimi-Sadjadi et al. (2001) investigated for satellite-based horizontal cloud categorizations. Furthermore, the proposed algorithm should be tested for other climate regions like the tropics with e.g. the Barbados Cloud Observatory (Stevens et al., 2016) or the Arctic region with the Ny-Alesund station (Nomokonova et al., 2017). The classification algorithm should be further extended by the aerosol and dust categories for models like COSMO-ART (Aerosols and Reactive Trace gases) which provide corresponding information. This would allow additional comparisons with the observations for these categories. An extension of the shown method is also desirable for comparisons with satellite-based cloud classifications like the one from DARDAR. The model cloud classification will be especially a very valuable product for comparisons with the upcoming EarthCare satellite, which will have all necessary Cloudnet sensors on board (Illingworth et al., 2015a).

Overall the presented verification techniques and results show the potential of the newly introduced cloud classification algorithm for the evaluation of clouds within atmospheric models. In addition, the generated cloud classification could be the base for additionally derived products like of the Cloudnet project.

*Code and data availability.* An example Matlab implementation of the cloud classification algorithm together with a sample COSMO-DE (April and May 2013) dataset is accessible at https://github.com/akiohansen/cloud-class-algorithm (last access: 12 October 2018). The DOI for the repository is https://doi.org/10.5281/zenodo.1458145 . The high-resolution ICON LES model output for the LACROS grid column for 26 April 2013 is available at the PANGAEA database: https://doi.pangaea.de/10.1594/PANGAEA.895303 . The Cloudnet observations of the LACROS supersite are available via SAMD: doi.org/10.17616/R3D944.

*Author contributions.* The cloud classification algorithm was jointly invented by all authors and implemented by Akio Hansen. All evaluation statistics and results were discussed together. The manuscript was written by AH with comments, ideas and suggestions by all the other authors. All authors have read and proved the final manuscript before submission.

*Competing interests.* None of the authors have any competing interests to declare regarding this study.

*Acknowledgements.* The authors thank the German Ministry for Education and Research for the funding of this study through the HD(CP)² project. The study is funded within the framework program "Research for Sustainable Development (FONA)", www.fona.de, under the number FKZ: 01LK1211A/C. The Cloudnet project, funded from the European Union's Horizon 2020 research and innovation programme under grant agreement No 654109, is acknowledged by the authors for providing the target classification produced by the algorithms of the University of Reading using measurements of LACROS. We are very grateful to Patric Seifert and the LACROS team for all their support and the Cloudnet measurements as well as for the Cloudnet products. We especially want to thank the German weather service (DWD) for sharing the COSMO-DE forecasts. We also wish to thank Ewan O'Connor for all his Cloudnet support. The authors acknowledge as well the huge work of the German Climate Computing Center (DKRZ) for delivering the ICON LES model data.

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
