# Peer review of "Model evaluation by a cloud classification based on multi-sensor observations"

_Geoscientific Model Development, 2018_

## Referee Comment (RC1) · Anonymous Referee #1 · 21 Dec 2018

**Model evaluation by a cloud classification based on multi-sensor observations** by Hansen et al. submitted to GMD.

**General comments and recommendation:**

This manuscript describes a hydrometeor classification algorithm that turns modeled thermodynamic profiles and profiles of hydrometeor concentrations into a hydrometeor classification, which can then be used to evaluate the models' ability to produce the right hydrometeor type at the right time and location. It is inspired by the CloudNet algorithm, in the sense that it uses the same hydrometeor categories, but otherwise is based solely on modeled output. In this case, it uses a given constant concentration threshold for all hydrometeors to decide which hydrometeor type dominates. In doing so, the authors make a useful, though simple, first step towards comparing hydrometeor types that are observed and modeled. This allows first glances into whether a model produces the right hydrometeors/clouds and can be used in time series, point to point comparisons or larger area comparisons, and the authors show a few of these for the COSMO operational weather model and the ICON LES model.

Despite this nice first effort, the authors do not convince (me) that this modeled classification provides more insight into the model's behavior than the available model output already can do. The authors state that modeled biases they find (for instance, the overestimation of ice in COSMO) can not be addressed using ice cloud fraction profiles, but do not show that their method works better. Because the classification is solely based on model output, and does not - unlike the title suggest - take into account hydrometeor aspects that are detected with multi-sensor information such as particle sizes and radiative effects - it cannot be used to make fair comparisons with observations. Instrument simulator techniques that have long been developed to address such issues are not discussed. The simple threshold for hydrometeor concentrations is not further tested or discussed, for instance, what this implies for a comparison with the instrument-based classification (by the CloudNet algorithm) where much more information about hydrometeors is available based on lidars and radars.

What surely does not help is that the manuscript suffers from poor writing, including poor argumentation, structuring, repetition, spelling/grammar mistakes and spoken language. Especially at the outset of the manuscript (the introduction and methods) the authors fail at clearly delineating what their new classification entails and implies.

My recommendation is to reject this manuscript, and ask the authors to work on their method in light of what has been happening in much more detail by the community, including the use of simulators, and present more convincingly and in detail how this provides detailed insight in how the model performs, more so than other methods or model diagnostics can do. I also ask the authors to considerably improve on the writing. Some specific comments to think about when preparing a new manuscript are included below.

**Specific comments:**

Title: This is a model evaluation by a cloud classification inspired by the CloudNet algorithm but based on modeled hydrometeor concentrations. The current title suggests that multi-sensor information is used in a detailed assessment of hydrometeor concentration type, size, radiative effects, which is not the case.

**Abstract:**

L3: an important step, not necessarily the first.

From the second paragraph: not immediately clear whether the classification is only applied to models, or if a revised version is applied to models and to observations.

L10/11: regarding different cloud types : unclear. You mean, the accuracy of forecasted cloud types?

Third paragraph: can the authors include results about how the classification works for COSMO/LES, and what such an evaluation has taught you about the vertical structure of cloud and cloud type in either model.

L13: as the observations. Which show detailed cloud structures, the obs or LES?

**Section 1. Introduction:**

The first two paragraphs are oddly written. You first make it sound like only a model variable is derived, but LWC is not specifically a model variable, it is a variable on its own, that can be derived from obs and models. Why is this bad? You mention evaluation of mixed-phase clouds, and that a modeled classification offers potential to investigate mixed-phase clouds in models. A classification can be used to evaluate ALL types of clouds - but the CloudNet classification may so far be only operational for supersites that are located in (midlatitude) regions where mixed-phase clouds are common. This is an important difference.
P2L10: atmospheric models like ..... now you should mention which models, and then cite literature.

P2L16: I disagree with your saying that the CloudNet products have not been used to evaluate the representation of clouds in models, in fact, you mention a whole list of them on L10. What you probably mean to say is that you can make qualitative comparisons between Cloudnet-derived products and models, and more precise quantitative (what you call direct) comparisons. The latter might have happened, but is not entirely fair, if the model does not have a comparable classification.

L17: a surrogate "offers"...

L19: Similar cloud classifications - what does similar here refer to?  Unclear.

L 23: rain radars do not see cloud droplets, only rain drops, so how can they provide detailed cloud information?

L26: most atmospheric models are numerical models. Are you making a classification that works on a certain range of model resolutions? Which ones? Is this applicable to everything from DNS to a GCM?

L26-33: this again is very oddly written, and the content of sentences is not logically connected. First you write that you make a new classification, so now we're expecting to hear more details on that. Then you write that standard metrics are not part of (a, your, which?) classification, ok fine, then what is? Then you talk about fuzzy methods ... is this what you use? Details on that this is inspired on methods

often used in the evaluation of precipitation is useful, and can be added, but really as a reader we are expecting to hear what is specific about your classification.

**Section 2. Data & methods**

P3L12/13: Remove "due to the remote sensing measurement characteristics"

A number of poor grammar/spelling here. The abundant use of e.g. throughout the manuscript is one example, including spoken language like: Isn't. Don't. Till. Spelling mistake: see occurence in the caption of Fig 4. This list is not comprehensive.

P4 second paragraph: Are we assumed to now understand how the classification works? Will you explain Fig 2? How are the thermodynamic profiles used? How does this connect to next paragraphs? Can you write it in a way that allows us to understand that what follows will be an explanation of Fig 2?

L25-35: which thresholds on QC/QI? You mention it later - include here?

P5L4-5: "would be thus difficult to interpret" poor grammar

P5L1: "have to be chosen" - but are in the end not chosen based on the instrumentation, but purely from a model point of view. For the lidars, the signals are certainly very sensitive to the concentration of particles, but for the radars, the signal will additionally be very sensitive to the particle sizes: a few large particles may already give you a return. Especially for cases where both ice crystals and cloud droplets are present, or where drizzle/rain drops and cloud droplets are present, the concentrations of the larger particles (crystals and rain drops) can give returns to the radar that would make the original CloudNet classification categorize this as a mixture of both, whereas the concentration of the larger particles might be small enough to let the model classification categorize this as just cloud droplets. COSMO only has a 1 moment scheme, and therefore you could not retrieve any information about the particle size, and perhaps this is why you have chosen just a simple threshold. But other models might have a 2 moment scheme, including the ICON-LES I believe, and because the lidar/radars are sensitive to both, this paper should at least have a discussion about how sensitive the choice of this threshold is.

What does a *significant* concentration mean for the model? How low/high is this concentration with respect to what we usually find in clouds.

P6L8-9: "by averaging the categories to the most frequent one": how does one average to a most frequent one? Don't you just select the most frequent one, no averaging involved? Is the most frequent one also the one that has the highest concentration? What if there is a warm cloud with one strong rain event, then the most frequent cloud category might be cloud droplets, but the highest concentration or strongest returns are at times of rainfall. My point is that such aspects are not discussed.

**Section 3:**

First sentence, first paragraph. It is an overstatement to say that your classification is consistent with the original CloudNet algorithm applied to observations. If anything, it is inspired by the CloudNet algorithm by selecting the same categories. The subsequent selection is purely based on modeled concentrations of different hydrometeors, which are in no way compared or comparable to lidar and radar retrieved signals that are functions of concentrations, particle sizes and more. The

authors also do not show how the original CloudNet algorithm does the selection, so there is no way for the reader to assess how much more detail the original classification entails.

PL26 : the temperature and humidity profile may be correct, but the microphysical scheme wrong. There is no way to separate the origin of these errors from a qualitative comparison of cloud categories. L29: detailed insights: give me an example.

P13L15/16: "Nevertheless, differences for example between the categories of "Drizzle or rain" and "Drizzle/rain & cloud droplets" can be identified, which provide detailed insights into the models' microphysics." Please provide the reader with what detailed insight you obtained regarding ICON's microphysical scheme. If you would give this to Axel Seifert, what does he learn from this what he could/did not know from other diagnostics.

P8L20: I am not at all convinced by your statement that identifying a modeled overestimation of ice clouds at the expense of clear skies is not feasible when looking at the mean cloud fraction profiles. Please show the ice fraction profiles of COSMO and the observations and explain why it does not reveal COSMO's bias.

P8L23-24: "Nevertheless, especially at high altitudes, the lower sensitivity of the remote-sensing instruments has to be considered which could reduce the observed frequency of occurrence of "Ice clouds". Isn't this exactly why you wish to have a modeled classification that more closely resembles what the instruments are seeing. Implementing instrument simulators in the model, and then doing the classification there, would be a fairer comparison wouldn't it. Now we are left wondering which one is right, and we cannot - unlike the authors state - identify the origin of errors and prove a detailed insight of why the model is wrong.

The first paragraphs of sections (point-to-point comparisons/fuzzy verification) repeat much of what has been written before, although they are written more clearly than before. The summary is certainly much clearer and better written that earlier parts of the text.

---

## Referee Comment (RC2) · Anonymous Referee #2 · 28 Dec 2018

This study presents a cloud classification product for numerical models, so that model output can be directly compared with observations via CloudNet. While I find this an interesting research question, I find that the current paper does not show sufficiently originality, either for the development of the technique or for the evaluation thereof. I unfortunately have to recommend rejection of the current paper, mainly for the following reasons:

1) While the authors state that this goes beyond a comparison of liquid/ice/etc. . . cloud fraction, in practice it remains close. I was expecting much more of a simulator approach here: Simulate the radar retrievals in the simulations, and classify clouds based on that. Such a project would have a more significant additional value on top of current apples to oranges comparisons.

2) The data set of the comparison with observations is very limited: Only two months over a single location. Not only results this in a much larger statistical uncertainty than necessary, it also means that the melting ice & cloud droplet category is not hit within the dataset.

3) The coarse graining algorithm ("take the most common category") does not seem all that accurate to me. Ideally, if CloudNet predicts some columns with (e.g.) liquid clouds and some separate columns with drizzle, the result on the model grid should be the mixture of the two.

4) Following up on point 3, it is unclear how the authors treat subgrid variability in the model. Are the subgrid parameterizations included at all in counting cloudiness? If so, does a pixel classify as liquid/ice/. . . if that phase is the dominant process, or is it possible to have multiple classifications within one model box?

---

## Author Comment (AC1) · 25 Jan 2019

Dear Reviewer #1, Dear Reviewer #2,

thank you very much for your detailed and comprehensive comments, ideas and recommendations on our proposed manuscript. These are very interesting and valuable for us to further improve our results, the method and the manuscript. Thank you!

We will try to incorporate all your suggestions, which are very helpful! Furthermore, we will advance our writing and the structure of the argumentation to increase comprehensibility. Another native speaker will be involved into our internal review process for the revision.

Please excuse the possible misleading title as we've seen by your expectations in your

reviews about the content. Therefore, we will change the title to a more precise one like: "Evaluation of clouds by a model-based cloud classification.".

Both reviews suggest the application of forward operators. Most certainly the usage of forward operators to create synthetic observations will account better for the instruments' characteristics. This approach will also be physically more consistent at the comparison of the model with observations. Nevertheless, we also think our proposed simple approach of a model based cloud classification has already an additional value for the evaluation of clouds. The approach shows the great potential of a synthetic cloud classification, which can be directly compared with Cloudnet observations. In addition, this easily accessible product provides a fast and comprehensive view on the accuracy of the modeled cloud type and profile. The results are very helpful as a starting point of a more detailed analysis.

On top of these reasons, the simple approach has great advantages compared with the forward operator approach. One example is its fast application on new model output and the easily understandable assumptions. The computation of all forward operators for the three required Cloudnet remote-sensing instruments (cloud radar, microwave radiometer, LiDAR) takes already a lot of time. The operators have to be further on fully consistent with the model's microphysics. This introduces at the same time another source of uncertainty to the analysis. All applied assumptions must be carefully considered at interpreting the results. Accordingly, the results of our simple approach are much better traceable and show clearly certain model issues.

For the above-stated reasons, we still think our proposed simple cloud classification algorithm is worthwhile to be used by the community. From our point of view, the reader should see the new cloud classification as described in reviewer #2 comments': "...the authors make a useful, though simple, first step towards comparing hydrometeor types that are observed and modeled. This allows first glances into whether a model produces the right hydrometeor/clouds and can be used in time series, point to point comparisons or larger area comparisons...". We see our cloud classification

algorithm as the first version of such an approach, which will be for sure enhanced like most geoscientific models. From our point of view, GMD is a journal about Geoscientific Model Developments and explains developments and tools in this field.

We agree with both reviewers that the forward simulator strategy is worthwhile to investigate, and we are already working on that. Nonetheless, the comparison of the complex approach with our simple approach will likewise be an interesting research question, which we want to investigate.

Comment Reviewer #2:

"2) The data set of the comparison with observations is very limited: Only two months over a single location. Not only results this in a much larger statistical uncertainty than necessary, it also means that the melting ice & cloud droplet category is not hit within the dataset."

The focus of our study is on the cloud classification algorithm and how to use the new classification for a detailed model evaluation. Therefore, the limited dataset of two months of operational COSMO-DE forecasts for a single location is only exemplary used to present the methods. The results themselves have for sure a large uncertainty because of this limited dataset, which we'll write explicitly in our manuscript.

So far our algorithm doesn't account for any subgrid variability. Only one single class is possible for each model box. Thank you very much for this valuable feedback and the ideas concerning the variability at the coarse graining, the chosen hydrometeor threshold as well as the subgrid variability. We will investigate the open questions in detail. Furthermore, we plan to perform sensitivity studies on the thresholds and the coarse graining. We will provide error estimates and uncertainties at the revision of our text.

We apologize very much if the text is not precise enough that our proposed simple cloud classification is only similar to the Cloudnet Target classification but not fully

consistent with it. Thank you very much reviewer #2 for sharing your impression with us. The suggested phrasing of "inspired by the CloudNet algorithm" is clear and easy to understand. We like this phrase very much and will include this into our text. Sorry for any misunderstanding regarding the Cloudnet Target classification. This synergetic product is one of the most comprehensive and most advanced multi-sensor products in our scientific field.

Thanks a lot for your further specific comments, ideas and findings. As explained before, we consider the following two suggestions, namely the usage of forward operators and the extension of the analyzed verification data set, to be beyond the scope and intention of our manuscript. Concerning all other suggestions and comments we fully agree with the two reviews. According to these issues we will be glad to revise our manuscript extensively if such an improved manuscript has the chance to be published by GMD. Therefore, we kindly ask for the possibility to revise our originally proposed manuscript.

Thank you very much once more for the reviews and thank you in advance for the chance to revise our manuscript,

Akio Hansen, Felix Ament, Andrea Lammert
* * *

---

## Author Comment (AC2) · 25 Jan 2019

Dear Reviewer #1,

thank you very much for your valuable and profound review of our proposed manuscript about our new model-based cloud classification. Your feedback is very helpful for us. You will find our detailed answer to your review as Author comment AC1 at the interactive discussion session.

Thanks a lot in advance and all the best,

Akio Hansen, Felix Ament, Andrea Lammert
* * *
[Figure]

2018.

---

## Author Comment (AC3) · 25 Jan 2019

Dear Reviewer #2,

thanks a lot for your very helpful suggestions and feedback on our proposed manuscript about our model based cloud classification. Your ideas and your review are very valuable for us. You can find our detailed answer to your review as Author comment AC1 at the interactive discussion session.

Thanks a lot in advance and best regards,

Akio Hansen, Felix Ament, Andrea Lammert
* * *
[Figure]

2018.